# Spatial exclusion leads to "tug-of-war" ecological dynamics between competing species within microchannels

**Jeremy Rothschild**[1], **Tianyi Ma**[1,2], **Joshua N. Milstein**[1,2]\*, **Anton Zilman**[1,3]\*

**1** Department of Physics, University of Toronto, Ontario, Canada, **2** Department of Chemical and Physical Sciences, University of Toronto Mississauga, Ontario, Canada, **3** Institute for Biomedical Engineering, University of Toronto, Ontario, Canada

\* josh.milstein@utoronto.ca (JNM); zilmana@physics.utoronto.ca (AZ)

**Data Availability Statement:** The code to run the analysis of the results can be found at https://github.com/jbRothschild/spatial-exclusion.

## Abstract

Competition is ubiquitous in microbial communities, shaping both their spatial and temporal structure and composition. Classical minimal models of competition, such as the Moran model, have been employed in ecology and evolutionary biology to understand the role of fixation and invasion in the maintenance of population diversity. Informed by recent experimental studies of cellular competition in confined spaces, we extend the Moran model to incorporate mechanical interactions between cells that divide within the limited space of a one-dimensional open microchannel. The model characterizes the skewed collective growth of the cells dividing within the channel, causing cells to be expelled at the channel ends. The results of this spatial exclusion model differ significantly from those of its classical well-mixed counterpart. The mean time to fixation of a species is greatly accelerated, scaling logarithmically, rather than algebraically, with the system size, and fixation/extinction probability sharply depends on the species' initial fractional abundance. By contrast, successful takeovers by invasive species, whether through mutation or immigration, are substantially less likely than in the Moran model. We also find that the spatial exclusion tends to attenuate the effects of fitness differences on the fixation times and probabilities. We find that these effects arise from the combination of the quasi-neutral "tug-of-war" diffusion dynamics of the inter-species boundary around an unstable equipoise point and the quasi-deterministic avalanche dynamics away from the fixed point. These results, which can be tested in microfluidic monolayer devices, have implications for the maintenance of species diversity in dense bacterial and cellular ecosystems where spatial exclusion is central to the competition, such as in organized biofilms or intestinal crypts.

## Author summary

Competition for territory between different species has far reaching consequences for the diversity and fate of bacterial and other ecological communities. In this study, we theoretically and computationally study the competitive dynamics of two bacterial populations competing for space in one-dimensional confined environments, extending classical

**Funding:** TM and JM acknowledge funding from the NSERC Discovery Grant RGPIN-2019-06520 (https://www.nserc-crsng.gc.ca/) and the New Frontiers in Research Fund Exploration Grant 207-2019-2020-Q4-00576. AZ and JR acknowledge support from the NSERC Discovery Grant RGPIN-2022-04909 (https://www.nserc-crsng.gc.ca/). The funders had no role in study design, data collection and analysis, decision to publish, or preparation of the manuscript.

**Competing interests:** The authors have declared that no competing interests exist.

models that serve as paradigms for competitive dynamics but do not explicitly include spatial exclusion. We find that spatial effects can drastically change the population outcomes by changing the probabilities that one species excludes the other from the environment. In comparison to the predictions of population models that neglect spatial aspects of the competition, species with higher relative fitness differences are less heavily favoured to outcompete their rival species. Spatial exclusion also changes the system stability with respect to invasion: there is a reduction in the effectiveness of the invader's ability to take over the population. Our results show that spatial exclusion has repercussions for species dominance and the long-time composition of populations. These must be considered when trying to understand complex bacterial ecosystems such as biofilms and intestinal flora.

## Introduction

Ecological competition is a ubiquitous feature of multi-species communities. It often manifests itself through direct antagonistic interactions between species, such as bacterial toxins, metabolic waste products and parasitic infections [1–3]. Competition also commonly occurs indirectly through various exploitative scenarios that deplete communal resources. Computational models of the dynamics of populations, framed in the context of a competition for finite system resources (e.g., light, food, population density, etc.) [4–12], have defined various heuristic measures of this competition for resources, such as the niche overlap, competition strength, and carrying capacity. Although these measures are commonly used to describe the dynamics of population growth and co-existence, a deeper understanding of the processes that govern the structure of ecological communities is acquired by exploring the mechanisms of the resource competition that underlie these coarse-grained, aggregate parameters [13–18].

Among the various resources required for population maintenance and growth, physical space is essential for expansion and access to additional resources [19–21]. In fact, individuals inherently require physical space for both their own growth and those of their progeny. Spatial competition can result in complicated patterning, synchrony of population distributions, spatial segregation into different niches within the environment and hosts, as well as other nontrivial dynamics [20, 22–25].

In bacterial communities, a variety of spatially ordered configurations may emerge from similarly distributed initial populations. This spatial structure plays an important role in medicine, industrial fabrication, and food production [26–30]. In bacterial biofilms, for instance, microbial populations form complex structures wherein various species segregate [31–33]. Different layers of bacterial species within a biofilm may have different sensitivity and resistance to antibiotics that restrict our ability to treat associated infections [34, 35]. The biogeography of bacteria in the digestive tract, which form the human digestive microbiome, illustrates another spatially heterogeneous ecology [36, 37]. In particular, the intestinal tract hosts diverse microbiota whose complex physical structures, such as mucus densities and epithelial crypts, have direct implications on the long-term composition of the bacterial community [38]. It is, therefore, necessary to understand how spatial constraints, arising from a confining environment and crowding/exclusion by other bacteria, shape the dynamics of each species and the overall patterning of the populations.

Spatial constraints may also have ramifications for the overall ecological diversity. For instance, the boundary between expanding fronts of different bacterial populations, grown on solid substrates, fluctuates superdiffusively. This encourages accelerated genetic drift that may

limit diversity more rapidly than neutral mutation models void of any spatial dynamics [39]. Alternatively, diversity may be increased in ecosystems wherein species undertake differing strategies in relation to the space they occupy—sometimes referred to as distinct spatial 'niches'. For instance, trade-offs between motility and competitive ability may allow for coexistence between competing species [40].

It is well known that diversity may be strongly influenced by the invasion of external species. As initially noted by Gause *et al.*, extinctions are frequently observed in a closed competitive ecosystem within a laboratory setting even though the similar ecosystem persists indefinitely in nature [41–44]. This suggests that invasion events, which implicitly rely on a partitioning of the space between the local and meta-community, contribute crucially to the population dynamics by reintroducing individuals into the ecosystem [45–48]. For instance, persistent diversity is observed through the fragmentation of continuous ecosystems in studies of patch-models of ecology and theories of island biogeography [16, 49–52].

The contest for space is critical in environments with small total populations, which accentuates the individual composition of the colony, such as within intestinal crypts [53, 54]. In recent years, microfluidic devices have started to provide controlled platforms for exploring the population dynamics of small bacterial microcolonies. Experiments in microfluidic monolayer devices, of various geometries, have shown that small populations of asymmetric bacteria, like *E. coli*, can align into highly ordered arrangements [55]. These populations behave differently from their well-mixed counterparts given the densely packed nature of their confinement [56, 57]. Cell morphology and the confining geometry are observed to greatly affect the ordering and fixation probability of cell populations in these devices [57–62].

Certain models, like the classical Moran model, subsume spatial factors into the well-mixed assumption that the species abundances are uniformly distributed in a certain location. Other models of ecological dynamics, such as island models, have incorporated aspects of special heterogeneity [63–65]. However, well-mixed continuous models that describe the concentration of bacterial populations generally neglect explicit spatial exclusion. Although many factors may influence competition for space, fundamentally cells in these densely packed populations must exclude each other through mechanical interactions. Consequently, modeling the competition of bacteria in small, confined environments requires explicit consideration of the system's spatial configuration to correctly describe the competitive dynamics.

We find that in confined environments, the spatial variation in species location may generate profound spatial disparities in reproductive opportunities. The collective division of cells engenders a phenomenon of skewed collective growth of the species domains, displacing a significant proportion of cells towards the peripheral boundaries, see Fig 1A. In this scenario, cells positioned in central regions of the space enjoy the advantage of prolonged reproductive success over several generations, while those situated at the environment's periphery face rapid extinction. This spatial variation in reproductive value carries significant consequences for the process of ecological dynamics. Over time, it promotes the emergence of long-lived lineages originating from the central cells, while lineages from cells at the ecosystem edges die out faster.

Previous studies have examined quantitative aspects of this spatial exclusion on population diversity. Inspired by the somatic evolution processes, [66] examined one-dimensional pushing dynamics of a linear a chain of cells growing out of a compartment with one exit, modelling a mother machine geometry. The study showed that spatial exclusion slowed down somatic evolution and delayed the onset of cancer within tissues of multicellular organisms. However, the linear pushing process studied was uni-directional, with cells leaving the system only from one end of the lane. As a result, the founder cells at the closed end of the pushing

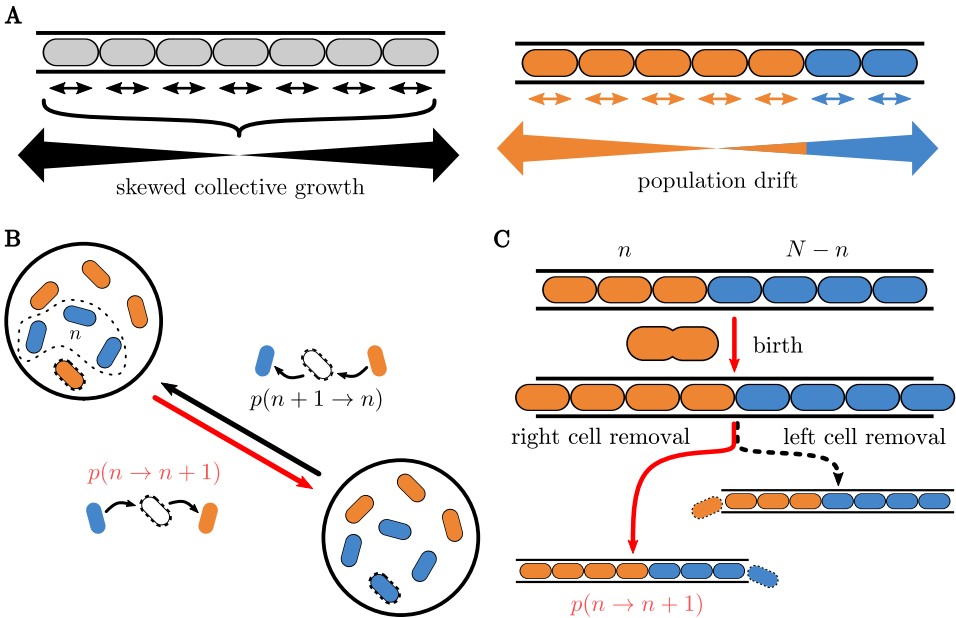

**Fig 1. Illustration of minimal models of ecological competition used in this paper. (A)** Schematic portrayal of the skewed collective growth. Left: dividing cells push on their neighbors which results in preferential movement of the cells toward the ends of the channel. Right: the skewed collective growth results in the asymmetric dynamics of the inter-species boundary. **(B)** The classical Moran model within a well-mixed population of two species of bacteria. Change in the abundance, *n*, of a species relies on selecting an individual of that species to give birth while simultaneously having another individual die. **(C)** A spatial exclusion model for 1-dimensional competition within a micro-channel. Instead of a well-mixed population, the cell populations are segregated, each to one side of the open channel; see text.

queue remained indefinitely in the process, and in the final fixated state, all cells were progeny of one founder cell.

Removing this physical constraint, a recent study explored the lineage diversity of a microbial population in a channel with two open ends [57], combining experiments and mathematical modeling. Focusing on the time of complete loss of lineage diversity in a system where at the initial configuration each cell belongs to a different species, the study revealed that lineage diversity can rapidly diminish within channels. However, the dynamics of fixation, extinction and invasion in a population of two species with fitness differences competing due to spatial exclusion in the one-dimensional space remain incompletely understood.

In this paper, we systematically explore how spatial constraints imposed by cellular interactions influence the population dynamics of competing cellular species, confirming and expanding previous work. Inspired by the experimental setting of a microfluidic chemostat, we investigate how two species of bacteria compete through physical exclusion in an open, single lane microchannel [57, 67].

The paper is structured as follows. We first review a general model of competing well-mixed populations—the Moran model—before describing an extension to this model that incorporates spatial exclusion. We calculate the probabilities and the mean first-passage times of fixation for this spatial exclusion model. We then explore how fitness differences between the two species—for instance, differences in the bacterial growth rate or doubling time—shape the competitive landscape by affecting the time to and probability of fixation. Finally, the

model is used to investigate invasion events that perturb a local community and the characteristics of a successful invasion.

## Models and methods

We characterize the state of the system by the abundances of the competing species (i.e. the numbers of individuals of each species found in the ecosystem) which reflect the dynamics and the evolution of the community. We focus on the processes where the competition amounts to a zero-sum game: different populations compete for dominance under a constraint of finite total population size, $N$, determined by limitations of the inhabited space. Thus, in a two species system with a finite total population size studied here, fixing the abundance of the population of the first species, $n$, determines the abundance of the population of the second species, $N - n$, and both species compete to maintain their non-zero abundances in the system. The constraint of a fixed population size means that the dynamics of this 2 species system can be mapped onto a one-dimensional process defined by the abundance $n$.

Several important models have been used to study the effects of different competition mechanisms on the species abundance and the community structures. A classical, highly influential model—the Moran model (and its variants)—has served as the paradigm for understanding the effects of stochastic ecological drift and natural selection on the diversity of a well-mixed population [59, 68–71]. A closely related model, Hubbell's neutral theory of biodiversity has been used to describe the emergence of the species abundance distribution in a neutral immigration-birth-death process [72]. Among others, Lotka-Voltera models further explore the role of species interactions and niche overlap on the interspecific competition; their frameworks can also be roughly mapped to the Moran model in their neutral regimes [16, 73, 74]. The fundamental stochastic nature of the ecological processes underlies all these models, where stochastic fluctuations of the abundances emerge from the demographic noise (i.e., the inherent randomness of birth and death events in a population).

We model the population dynamics of two competing species as a discrete stochastic process denoting the probability of being in a state with one species abundance at $n$ (and the other species abundance at $N - n$) at time $t$ is $p(n, t)$. The population abundance of a species can change either through births or deaths of the individual cells, with the probability of a birth or death in the population $n$ in an interval of time $\Delta t$ denoted as $T^+(n \rightarrow n + 1, \Delta t)$ or $T^-(n \rightarrow n - 1, \Delta t)$, respectively [75, 76]. The evolution of the probability $p(n, t)$, is governed by a one-dimensional forward master equation (ME)

$$\partial_t p(n, t) = r_N^-(n+1)p(n+1, t) + r_N^+(n-1)p(n-1, t) - \left(r_N^+(n) + r_N^-(n)\right)p(n, t), \quad (1)$$

where $r_N^\pm(n) = T^\pm(n \rightarrow n \pm 1, \Delta t)/\Delta t$ are the transition rates for events of an increase or decrease in abundance [77].

We are interested in the process of fixation wherein the abundance of one species approaches $N$, effectively outcompeting the other species by removing it from the system. This fixation can be viewed as a first-passage process that occurs when the abundance of a species reaches either of the absorbing states, at $n = 0$ and $n = N$, at which point the system settles at steady-state with one species dominating indefinitely [16, 78]. In these processes, the first-passage probability and the mean first-passage time (MFPT) are characteristics of the system which elucidate the dynamics of the process [76].

The mean-first passage time to either fixation state, from a starting abundance $n$, $\tau(n)$, relates the average time the competition between the two species lasts before one takes over

and is described by the backward equation

$$
\begin{aligned}
\tau(n) \quad &= \Delta t + T^+(n \to n+1, \Delta t)\tau(n+1) + T^-(n \to n-1, \Delta t)\tau(n-1) \\
&\quad + (1 - T^+(n \to n+1, \Delta t) - T^-(n \to n-1, \Delta t))\tau(n).
\end{aligned}
\tag{2}
$$

The master Eqs 1 and 2 can be solved numerically to obtain the probability and MFPT of fixation, (see Section A in S1 Appendix). Alternatively, the Fokker-Planck (FP) expansion to order $\mathcal{O}(N^{-2})$ of the backward equation is

$$
-1 = \frac{A(f)}{N}\tau'(f) + \frac{B(f)}{2N^2}\tau''(f),
\tag{3}
$$

where the drift and diffusion terms are defined as $A(f) = r^+(f) - r^-(f)$ and $B(f) = r^+(f) + r^-(f)$ with the transformation $r_N^{+/-}(n) \to r^{+/-}(f)$ and $f = n/N$. A FP expansion in $1/N$ of the probability of the first passage to settle in either absorbing abundance states $F \in \{0, 1\}$ results in a continuous description (see Section B in S1 Appendix)

$$
0 = A(f)P_F'(f) + \frac{B(f)}{2N}P_F''(f),
\tag{4}
$$

where $P_F(f)$ is the probability of being absorbed at the state $F$ starting from an abundance $f$. For instance, $P_1(f)$ is the probability that a species with fractional abundance $f$ will fill the space to fixate at the absorbing abundance $F = 1$. An equation can also be derived for the MFPT conditioned on the fixation to an absorbing state. The discrete form of these equations all have exact numerical solutions [79] (see Section C in S1 Appendix).

The solutions to these equations depend on the choice of the birth and death rates of the model. In the classical Moran model, which represents mixed populations without spatial structure, a random individual from a fixed and finite population of size $N$ is selected to give birth at each time step while, simultaneously, a random individual is selected to be removed from the system to make room for the progeny and maintain a constant total population $N$ (see Fig 1A and Section C in S1 Appendix).

Building on this paradigmatic model, we consider a system where $N$ individuals of two species are constrained to a one-dimensional space (a channel open at both ends) as shown in Fig 1B. We assume that the channel is always full and that the two species are segregated such that only one boundary separates the populations. Without loss of generality, we take $n$ to describe the number of individuals belonging to the species on the left side of the lane (species 1). Contrary to the Moran model, the transition rates of the populations now depend on the spatial arrangement of the cells. A cell at any location can divide and produce a progeny, but death events only happen when a cell is pushed out of either end of the channel.

In this system, the relative species abundance (frequency) delineates the location of the boundary that separates the species. When an individual cell grows, the boundary shifts right or left, as illustrated in Fig 1C. As $n$ increases, the boundary between the two species moves to the right with probability $T^+(n \to n+1)$; conversely, decreases in $n$ result in the boundary moving to the left with probability $T^-(n \to n-1)$. This competitive process continues until a sequence of jumps makes the boundary reach either end of the channel (i.e., $n = 0$ or $n = N$) with only one surviving species, which is said to fixate.

Like the Moran model, fitness differences modify the selection probabilities. The probability of selecting a specific individual $i$ from species 1 for birth with relative fitness difference $w$ —related to the selection parameter $s$ commonly used in evolutionary dynamics as $w = 1 + s$— is $p_{i,1}^+ = w/[(N - n) + wn]$, whereas the probability for individual $i$ of species 2 is $p_{i,2}^+ = 1/[(N - n) + wn]$ [80, 81]. Subsequent to the birth, the offspring must create space for itself

by pushing upon the adjacent cells situated on either side of its progenitor. This triggers a cascade of coordinated cellular displacements along the channel, culminating in the expulsion of a terminal cell from the microchannel. This skewed collective growth causes the population boundary to preferentially drift towards an edge of the channel, eventually expelling one of the two species and fixating the other.

In a recent experiment in open-ended micro-channels, it was observed that cells are more likely to grow in the direction of the closer channel opening because fewer cells need to be pushed in that direction. As a first approximation, the probability for a cell to grow towards one of the openings can be expected to scale linearly with the number of individuals between the cell location and the other opening, as was indeed observed in [57]. The magnitude of this bias towards an end of the channel has been estimated in this particular experimental system [57]; however there are as of yet no systematic estimates for different experimental conditions. In this work, we make the minimal assumption that the probability that an individual cell at position $i$ divides to the right is proportional to the number of individuals that are to the left of it $p_{\text{right}} = (i - 1)/(N - 1)$ and vice versa $p_{\text{left}} = (N - i)/(N - 1)$, see also [57] and SI. This choice corresponds to a somewhat larger bias than estimated and modelled in [57] and the effects of changing this assumption will be investigated in the future. However, the effects of varying this bias on the investigated quantities (invasion probabilities) appear to be minimal [57] (See also Discussion). This spatial exclusion model is illustrated in Fig 1C.

Thus, the rate at which the boundary moves to the right in an interval of time $\Delta t$ is the sum of the rates for each individual of species 1 to grow to the right

$$r_N^+(n) = \sum_{i=1}^{n} \frac{1}{\Delta t} \left( \frac{w}{(N - n) + wn} \right) \frac{i - 1}{N - 1} = r \frac{wn(n - 1)}{2[(N - n) + wn]}, \tag{5}$$

whereas the rate of the boundary moving to the left is

$$r_N^-(n) = \sum_{i=n+1}^{N} \frac{1}{\Delta t} \left( \frac{1}{(N - n) + wn} \right) \frac{N - i}{N - 1} = r \frac{(N - n)(N - n - 1)}{2[(N - n) + wn]}. \tag{6}$$

Here, we have rescaled the basal rate to $r = 1/(N - 1)\Delta t$. The summation of individual cell growth rates contributes to the cumulative growth of the entire cell population, thereby influencing the skewed directional movement towards either extremity of the channel. These rates can be substituted in Eqs 3 and 4 and solved for the dynamics of the spatial exclusion model.

## Results

### Spatial exclusion gives rise to sharp sigmoidal fixation probabilities and exponentially fast MFPTs

We first consider a neutral case where species are functionally equivalent without fitness differences between them ($w = 1$) (e.g., two populations of cells whose phenotypic differences offer no upper hand or two identical lineages with a common ancestor). In contrast to the neutral Moran model where the drift term is zero, the drift term in the spatial exclusion model is strictly positive: $A(f) = N[(w - 1)f^2 + 2f - 1]/2[1 + f(w - 1)]$. Although an analytical solution to Eq 4 is not available for $w \neq 1$, the equation can be easily numerically integrated (see also Section A in S1 Appendix). Fig 2 shows the results of a comparison between the neutral Moran model and the spatial exclusion model.

We find that the probability for a species to fixate in the spatial exclusion model as a function of its initial fractional abundance is a sigmoidal function. This differs significantly from the linear dependence predicted by the Moran model (see Fig 2A). Any minority population

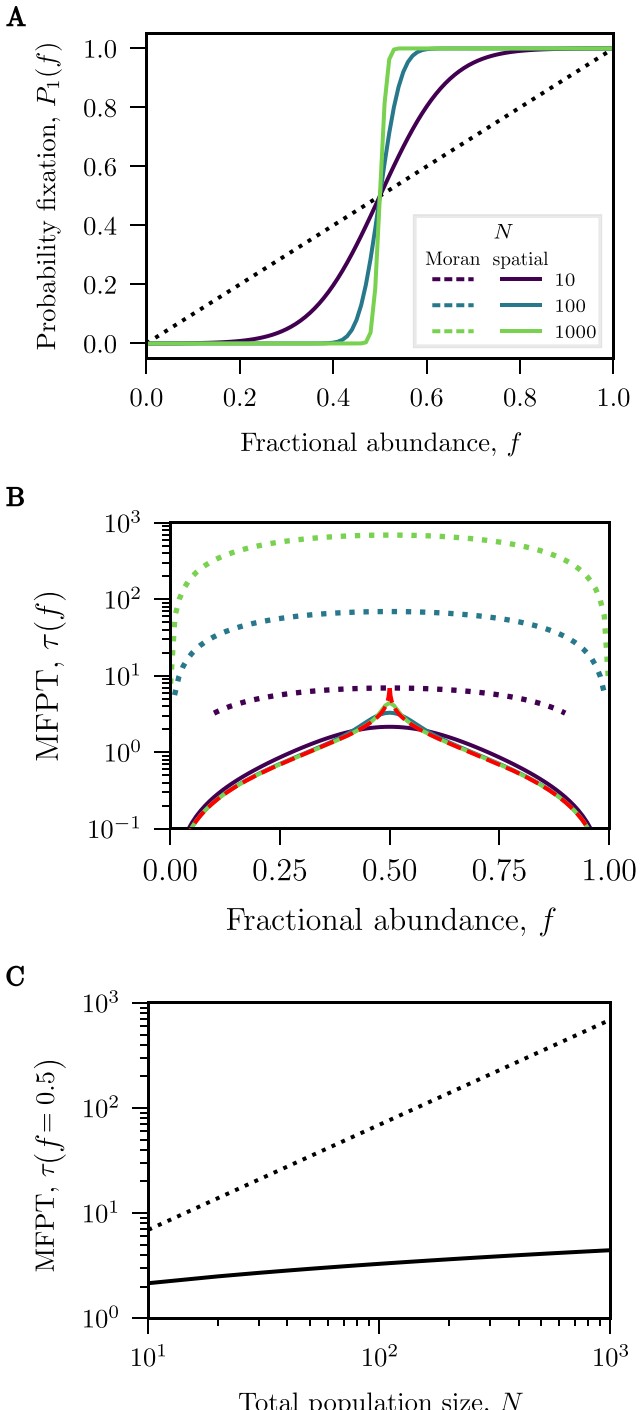

**Fig 2. Competition outcomes for neutral dynamics.** Fokker-Planck approximation for the fixation probabilities and mean-first passage times (MFPTs) for the Moran model (dotted lines) and the spatial exclusion model (solid lines) for various system sizes ($N$ = 10, 100, 1000). **(A)** The fixation probability in the spatial model is sigmoidal around an inflection point at $f = 1/2$ and becomes increasingly steep, approaching a step-function at large $N$. The Moran model, in contrast, always predicts a linear fixation probability equal to the fractional abundance. **(B)** For the spatial exclusion model, the MFPTs at varying $N$ collapse onto each other for most fractional abundances except around the inflection point. The dynamics away from $f = 1/2$ follow a deterministic path to fixation approximated by $\tau_{det}$ (dashed red line) from Eq 7. Notably, the MFPTs in the spatial model are significantly faster than those predicted by the Moran model. **(C)** The maximal MFPT of the Moran model is linear in $N$ whereas the maximal MFPT of the spatial model grows substantially slower and sub-linear in $N$; see Fig 3C.

(with a starting fraction $f < 1/2$), is much less likely to take over the population than in the Moran model. Conversely, any majority population ($f > 1/2$) is much more likely to succeed at fixating within the lane. The slope of this sigmoidal probability depends on the length of the microchannel (or the total population $N$) approaching a step function for large $N$. The inflection point is found at the equiprobable takeover abundance $f_{eq} = 1/2$, defined as the abundance at which both species are equally likely to take over the channel.

For the Moran model, the mean time for either species to fixate (the unconditional fixation time) grows linearly with the system size. For the spatial exclusion model, the mean time to fixation is much shorter than that of the Moran model for all initial boundary positions as shown in Fig 2B) (for the MFPT conditioned on the success of one of the two species, see Section D in S1 Appendix). Interestingly, the MFPT curves for different $N$ collapse onto each other away from the central peak at the equiprobable takeover abundance of 1/2. Thus, the time to fixation does not depend on system size unless the initial fractions of the two populations are closely balanced, in which case it is approximately logarithmic in $N$ as discussed below.

This independence of the dynamics on $N$ can be heuristically explained by investigating the behaviour of Eq 3. For the spatial model, the first term on the right-hand side (the drift term) is small, $A(f) \approx 0$, around the peak of the MFPT. The dynamics around the peak are dominated by the second (diffusion) term on the right-hand side of Eq 3, which scales like $1/N$. Conversely, away from the peak, the drift term dominates the expression for large $N$. In this case, Eq 3 can be approximated by a first-order ODE

$$-1 = \frac{A(f)}{N} \tau'_{\det}(f). \tag{7}$$

The solution to this equation is the time to fixation for a process with a deterministic velocity $A(f)$,

$$\tau_{\det}(f) = -\frac{1}{r} \left( \log|(w - 1)f^2 + 2f - 1| - \Theta(f_{eq}) \log(w) \right), \tag{8}$$

where $\Theta$ is the Heaviside function. Given that the deterministic velocity is proportional to $N$, the fixation is independent of $N$. The displacement of the fractional abundance is exponential in time when the change in the fractional abundance is governed by this deterministic velocity.

The mean time to fixation of the spatial exclusion model substantially differs from the prediction of the Moran model, even close to the peak, where the MFPT depends on $N$. This difference between the two models for the MFPTs to fixation is most apparent at a starting fraction of $f = 1/2$ where $A(f) = 0$. In Fig 2C, the MFPT to fixation in the neutral spatial exclusion model grows only sub-polynomially with $N$ rather than linearly as in the classical Moran model. We return to the quantification of this sub-polynomial growth in $N$ below.

## Fitness differences break the symmetry of the fixation probability and engender longer MFPTs

Fig 2 presents results of unconditional fixation times and probabilities of fixation for neutral populations with different but functionally identical species. However, more generally, phenotypically dissimilar species exhibit differences in their dynamics such as growth and death rates, efficiency of resource consumption, etc., which may impact their overall fitness in the environment. In this section, we present the results for the population fixation in the presence of a relative fitness difference, $w$, between two species, shown in Fig 3 for $w = 1, 1.5, 10, 100$. Very small relative fitness differences (such as $w = 1.1$) do not significantly alter the probability and MFPT to fixation compared to the neutral model as shown in Section D in S1 Appendix.

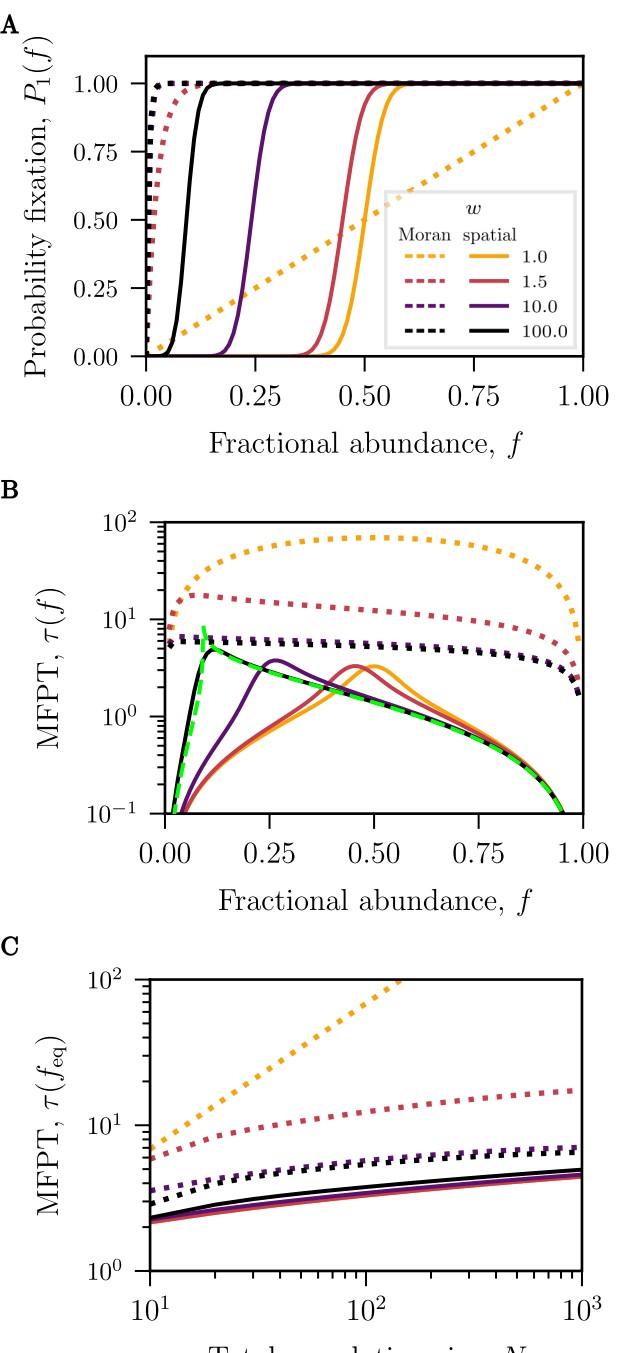

**Fig 3. Competition outcomes for dynamics with fitness differences.** Fokker-Planck approximation for the fixation probabilities and the mean-first passage times (MFPTs) for both the Moran model (dotted lines) and the spatial exclusion model (solid lines) for the values of relative fitness difference $w = 1, 1.5, 10, 100$. The fractional abundance $f$ is the abundance of the species with higher fitness. **(A)** Fitness difference increases the fixation probability of the fitter species, shifting the fixation probabilities curves towards lower abundances compared to the neutral case; $N = 100$. Note that the probabilities of fixation for the Moran model with $w = 10$ and $w = 100$ are effectively indistinguishable. **(B)** In the spatial model the maximum MFPT slowly increases with fitness in contrast to the Moran model where the MFPTs tend to diminish with $w$. For the spatial model, the dynamics outside a narrow region around $f_{eq} = 1/(1 + \sqrt{w})$ follow a quasi-deterministic path to fixation approximated by $\tau_{det}$, (shown in a dashed green line for $w = 100$) obtained from Eq 8. **(C)** The first passage time for the neutral ($w = 1$) Moran model exhibits linear growth in $N$ which slows down for higher $w$. For the spatial model, the maximal fixation times (MFPTs) scale sublinearly with $N$, with minimal effect of the fitness difference. See Fig 4 for further discussion.

Although results are shown for $w > 1$, results for fitness differences $w' < 1$ can be obtained by interchanging species 1 and 2 with $w = 1/w'$.

In the Moran model, the probability of fixation is strongly favored towards the species with a fitness advantage for even modest values of $w$, (Fig 3A), with the probability of fixation being $p(f) = (1 - w^{-fN})/(1 - w^{-N})$ for the fitter species [82]. Selection quickly skews the fixation probabilities so that the species with the advantage almost always fixates regardless of its initial abundance.

On the contrary, similar fitness differences do not influence population dynamics as markedly in the spatial exclusion model. Whereas the probability of fixation as a function of the initial fraction $f$ changes drastically in the Moran model, from a linear function at $w = 1$ to a concave function without an inflection point for $w > 1$, within the spatial model, the shape of these curves remains relatively unchanged for all values of $w$ (see Fig 3A). Rather, fitness differences shift the inflection point of the sigmoidal fixation probability curves towards lower initial fractional abundances $f_{eq} = 1/(1 + \sqrt{w})$, but a species is still significantly favoured to fixate when its fractional abundance is above the inflection point. The fact that the effect of fitness difference is blunted by the spatial competition is also reflected in the probability of fixation averaged over all initial abundances, see Section E in S1 Appendix: the average probability of fixation for the Moran model is higher than the average probability in the spatial exclusion model.

Fitness advantages also impact the dynamics of fixation. As shown in Fig 3B, higher fitness differences in the Moran model reduce the mean fixation time (unconditioned on the success of a species). In the spatial exclusion model, although a greater fitness leads to faster fixation conditioned on the species success (see Section D in S1 Appendix), this is not always the case when considering the unconditional mean fixation time. The dependence of the unconditional mean fixation time on the relative fitness in the spatial model is non-monotonic and location dependent as shown in Fig 3B. In particular, the mean fixation time is maximal for populations initialized at abundances close to the equiprobable takeover abundance $f_{eq} = 1/(1 + \sqrt{w})$, $\tau$ ($f_{eq}$) for any fitness (see Fig 3B). The fixation time marginally increases with the fitness difference between the species scaling logarithmically with $1/(w - 1)$ as follows from Eq 8. For this reason, the times to fixation for various fitness differences almost collapse onto each other for all system sizes $N$, as shown in Fig 3C.

Accordingly, in the spatial exclusion model, the longest timescale of the system is determined by the competition dynamics around $f_{eq}$ where the probabilities of either species successfully taking over are roughly equivalent. The asymptotic scaling with the system size $N$ of this maximal MFPT in the presence of fitness differences is similar to the asymptotic scaling in the neutral case (see Fig 3C)—much slower than the Moran time and sublinear with $N$, as further explored in discussion around Fig 4.

For initial conditions outside the close vicinity of the equiprobable abundance $f_{eq}$ fixation follows a quasi-deterministic course where the fixation times are almost independent of the system size for any initial condition. The deterministic approximation to the fixation time calculated from Eq 8 is shown in green dashed line in Fig 3B and serves as an excellent approximation except in the close viscinity of $f_{eq}$.

In the close viscinity of $f_{eq}$ species abundance dynamics correspond to a "tug-of-war"; the two species abundances fluctuate around the equiprobable takeover abundance, with both species trying to take control of the channel by their abundance. Once the abundance diffuses away from the equiprobable point and one species becomes dominant, the dynamics quickly switches to a deterministic trajectory where skewed collective growth further accelerates the growth of the dominant species abundance, causing the rapid collapse in the abundance of the other species.

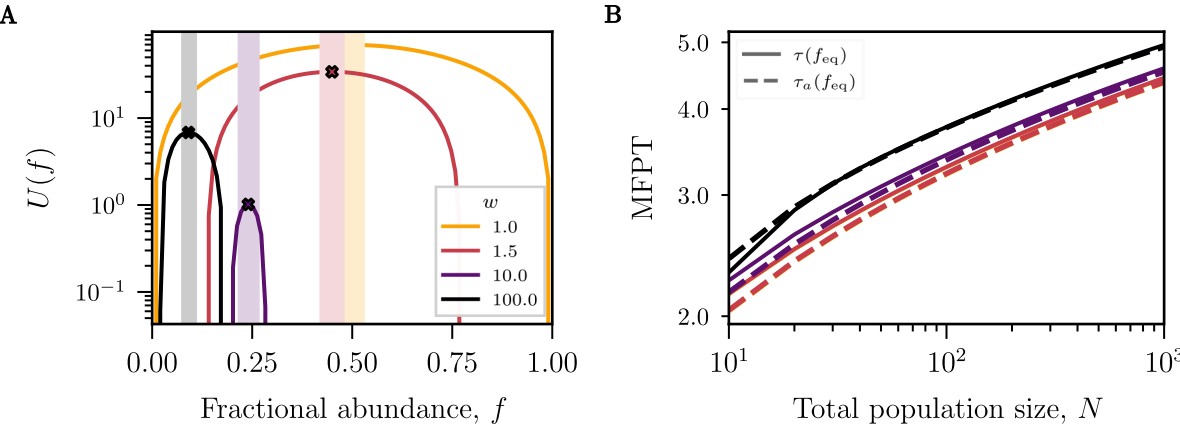

**Fig 4. Heuristic approximation of the MFPT and scaling with N. (A)** Graphs of the potential $U(f)$ with the location of the maximum, $f_{eq} = 1/(1 + \sqrt{w})$, indicated at the peak of the curves. The shaded areas represent regions along the channel, $f$, where the stochastic behaviour heuristically dominates the dynamics ($N = 100$). **(B)** The maximal mean fixation time (solid lines) as a function of system size increases with fitness advantage and scales approximately logratihmically with system size $N$. The quasi-deterministic approximation $\tau_a(f_{eq})$ (dashed lines) is very close to the exact dynamics. The maximal MFPT for $w = 1.0$ and $w = 1.5$ overlap at this scale. Colour coding for both panels is presented in the legend of panel A.

## Size dependence of the fixation times in different dynamical regimes in large systems competing in a tug-of-war

The dependence of the fixation time of any species on the size of the system $N$ is of crucial importance for the the speed of diversity loss, and therefore for the inference from experimental observations that commonly probe only the transient composition of evolving ecosystems. In Fig 3C we have numerically seen that the asymptotic behaviour of the MFPT to fixation grows sublinearly in $N$ for the spatial exclusion model (see Section D in S1 Appendix). In this section, we investigate further the large $N$ scaling of MFPTs—a regime relevant for many experimental systems that are often comprised of large numbers of individuals such as crevices in biological systems (e.g. gastrointestinal crypts [83–85]) or narrow channels in which cells navigate their environment (e.g. in Antarctic icesheets [86, 87]). We note that the results closely follow the asymptotic scaling already for $N \sim 100$, on the order of the typical number of bacteria in microfluidic channels and mother machines.

The one-dimensional Fokker-Planck equation for the MFPT to fixation, Eq 3, may be rewritten as

$$-1 = \frac{B(f)e^{U(f)}}{2N^2} \partial_f \left( e^{-U(f)} \partial_f \tau \right), \tag{9}$$

where

$$U(f) = -2N \int_f dy \frac{A(y)}{B(y)}. \tag{10}$$

$U(f)$, sometimes referred to as the Fokker-Planck potential, describes an effective potential landscape in which the boundary between the two species moves (see Section B in S1 Appendix). A general compact integral form of the solution to Eq 3, which is shown in Section G in S1 Appendix, can be evaluated for different potentials representing different population dynamics.

For the model defined in Eqs 5 and 6, $U(f)$ is a unimodal distribution with an unstable maximum found at $f_{eq} = 1/(1 + \sqrt{w})$ for $w \geq 1$ (see Fig 4A). Thus, unlike the more familiar problem of calculating the MFPT to cross over a potential barrier (Kramers Theory) [76], here we are interested in finding the MFPT to descend from a potential peak starting at an unstable point. This distinction means that the saddle-point approximation—which is commonly employed for the evaluation of asymptotic of the MFPTs shown in Eq 3—is inadequate in this case.

To derive an approximation for the asymptotic behaviour of the fixation MFPT, we heuristically separate the space into regions of predominantly deterministic or stochastic dynamics. As shown in Section G in S1 Appendix, the boundaries between these two regions naturally emerge from the integral form of the MFPT as the solutions to the equation

$$|U'(f_t)| \simeq \sqrt{U''(f_{eq})/\pi}, \tag{11}$$

and are depicted in Fig 4A. The interval $X_t = [f_t^-, f_t^+]$ is the region around $f_{eq}$ where the dynamics are dominated by stochastic diffusion with the frequency-dependent diffusion coefficient $B(f)$ arising from a "tug-of-war" dynamics between the two species. Conversely, outside of this region, the dynamics are well approximated as deterministic and are dominated by an ecological drift with a frequency-dependent velocity $A(f)$ arising from the skewed collective growth towards the channel exits.

Thus, if the initial fractional abundance of species 1 lies in the deterministic region $x \notin X_t$, the time to fixation is well approximated by the deterministic time $\tau_{det}(f)$ of Eq 8. On the other hand, if the initial fractional abundance is in the stochastic region $x \in X_t$, its motion is dominated at first by diffusion until it reaches one of the boundaries within a time $\tau_{dif}(f, f_t^{+/-})$, after which it typically continues along a deterministic trajectory. For illustrative purposes, we examine the fixation time starting at $f = f_{eq}$ given that there is an equal probability to diffuse to either boundary from this position. Thus the maximal MFPT can be approximated as

$$\tau_a(f_{eq}) \simeq \tau_{det}(f^\pm) + \tau_{dif}(f_{eq}, f_t^\pm) \sim \frac{1}{2} \log(\pi N) + \tau_{dif}(f_{eq}, f_t^\pm), \tag{12}$$

which scales as $\mathcal{O}(\log N)$ for large $N$, as is shown in Fig 4B (see also Section G in S1 Appendix). As shown in Section G in S1 Appendix, numerically $\tau_{dif} \approx 0.4$ and is independent of $N$ or the relative fitness difference; further analysis is needed to determine the constant analytically. Consequently, the longest timescale observed for species to fixate in the system increases slowly with the length of the channel so that rapid fixation may occur even in very long channels.

## Invasions in spatial models are less likely to succeed but succeed on shorter timescales on average than in well-mixed models

Once one species has fixated, it remains dominant in the channel unless an external event, such as an invasion or mutation, perturbs the system by introducing a new species variant that could compete against the established strain [88–90]. In microchannels, a mutation event could introduce a new variant at any location, but immigration is normally possible only from the edges. However, bacterial populations that grow in wider microchannels have been shown to organize into parallel lanes aligning and growing along the axis of the channels. These aligned lanes permit rare immigration events from one lane into another previously fixated lane, which can be viewed as the invasion of a new species into a lane [57, 91].

We are interested in the probability and the mean time of a successful invasion, wherein an individual cell of the invading species is inserted anywhere in the channel. Success of an invasion is defined as the invading species taking over and fixating within the lane/channel. Whereas the MFPTs presented in previous sections were the fixation times for any species to fixate, the invasion fixation time refers to the MFPT conditioned on the success of the invader. In the well-mixed Moran model, when an individual from a new species invades a previously fixated system, the fractional abundance of the invading species is $1/N$. As derived in earlier sections, the probabilities of a successful invasion fixation for the Moran model ($P_1(1/N) = (1 + w^{-1})/(1 + w^{-N})$) are depicted in Figs 2 and 3.

In contrast to earlier sections, the appearance of an invader in the one-lane model results in two rather than one inter-species boundaries as illustrated in Fig 5A. Nevertheless, the mathematical framework of competitive spatial exclusion described above allows us to model interspecies dynamics after such an invasion event, as shown in Fig 5A. The rules that determine

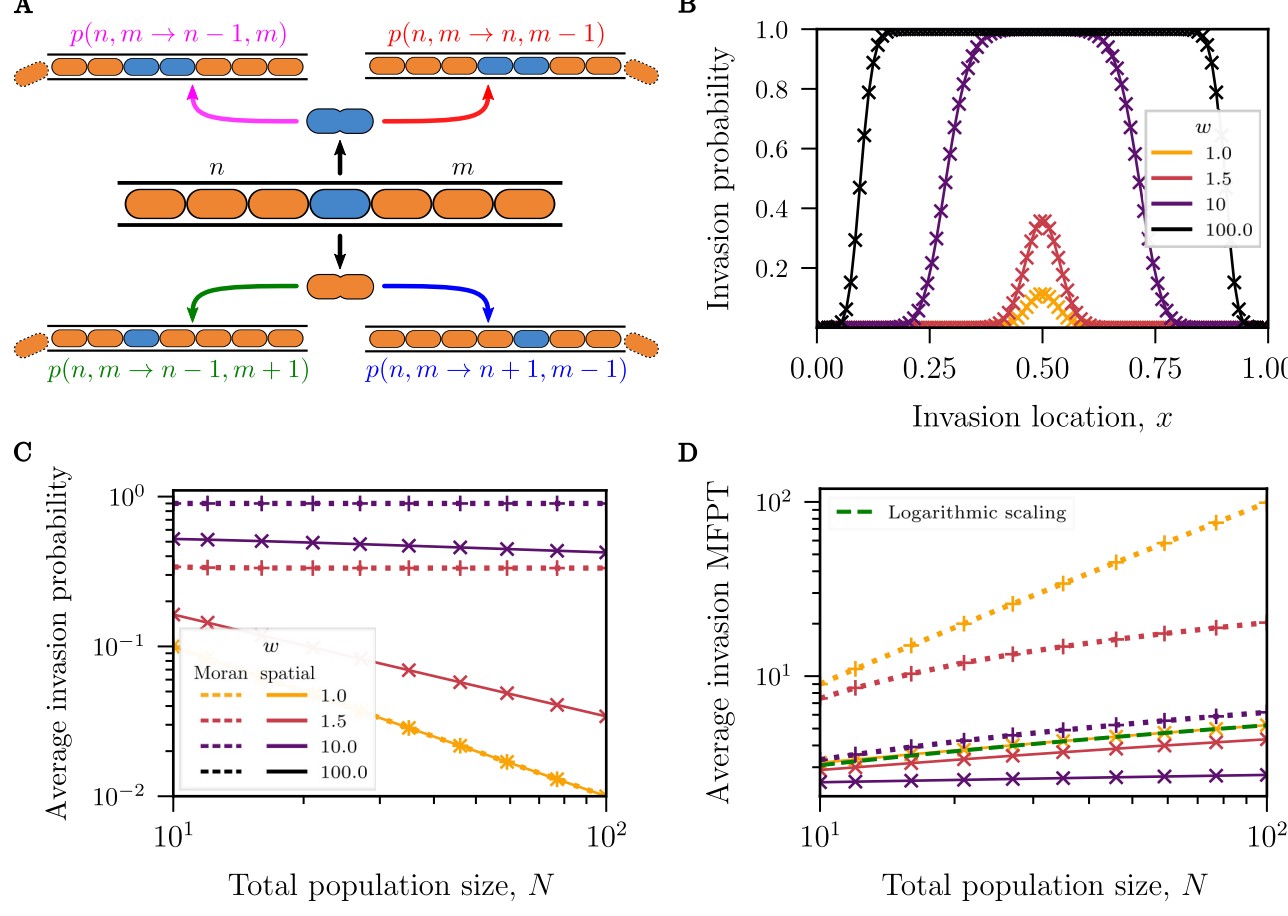

**Fig 5. Invasion into a fixated channel. (A)** Illustration of the invasion/mutation model wherein an invader (blue) with relative fitness difference $w$ infiltrates a native population (orange). The invasion is shown at the center for illustrative purposes. **(B, C, D)**: Numerical solutions to the Master equation for the spatial model (solid lines with × symbols) and Moran model (dotted lines with + symbols). **(B)** The probability of a successful invasion as a function of the insertion location ($N = 100$). Invasion events at the center of the channel are the most likely to succeed, but a relative fitness difference by the invading species increases the range of likely successful invasions. **(C)** The average probability of a successful invasion obtained by averaging with uniform weight the probability of successful invasion at all locations along the micro-channel. For $w = 1$, both models show similar behaviour. **(D)** The average mean time of a successful invasion in the spatial model increases with the population size, but less rapidly than in the Moran model. As the relative fitness difference increases, the mean time of successful invasions averaged over all invasion events decreases as well. Logarithmic dependence on $N$ of the complete lineage fixation time derived in [57] is shown as the green dashed line.

the probability for a cell to be selected for birth and death are identical to those outlined previously. However, the system defined by the two inter-species boundaries is now two-dimensional and, accordingly, has added states with rates defining additional transitions. The expressions for these rates and the corresponding two variable master equation are outlined in Section F in S1 Appendix. Given that the state space for the spatial invasion model is now two-dimensional, we rely on numerical solutions for the discrete dynamics instead of solving the corresponding 2D Fokker-Planck equation.

The outcome of an invasion in the spatial exclusion model is strongly dependent on the initial location of the invasion event, $x$. We find that invasions starting about the center of the channel are the most likely to succeed in pushing out the established species, see Fig 5B; this has previously been shown in the neutral case [57]. The probability that an invasion event succeeds increases with the fitness advantage of the invading species and successful invasions occur over a broader range of locations along the channel.

To provide a global measure of invasion success, we calculate the average invasion probability over all initial conditions. Notably, the average invasion probability in the neutral spatial model($w = 1$) is identical to the invasion probability of the neutral Moran model and is equal to $1/N$ as shown in Fig 5C. Intuitively, since all cells are identical and an invasion can arise at any location, one of the $N$ cells will become the ancestor of the entire population. This implies that the average probability of a successful invasion is $1/N$.

As demonstrated in Fig 5C, the probability of a successful invasion increases with fitness more sharply for the Moran model than for the spatial model. However, the probability of successful invasion remains low even for an invasive species with a ten-fold increase in fitness advantage, which is still less likely to fixate than the native species. This suggests that the spatial exclusion dynamics modeled here limit the competitive edge of strains with fitness advantages.

Although the average probability for the invasive species to fixate is identical under neutral conditions for both the Moran model and the spatial exclusion model, the dynamics of fixation differ between the two models. A global measure of the invasion dynamics is the average MFPT of a successful invasion, weighted by the probability of success at each initial invasion location, shown in Fig 5D (see Section B in S1 Appendix for details on conditional MFPTs). In the neutral case, this time is equivalent to the mean time to the complete loss of lineage diversity considered in [57].

The conditional MFPT of a successful invasion is larger for the Moran model than the spatial model in the neutral case. This is also true for an invader when fitness differences are incorporated: successful invasions fixate more rapidly in the presence of spatial exclusion than in the well-mixed model.

We find that the average conditional MFPT of a successful invasion in the spatial model changes non-monotonically with fitness advantage in Fig 5D. At low fitness differences ($w \leq 10$), the conditional MFPT decreases when fitness differences increase (see Section B in S1 Appendix); intuitively, an increase in fitness difference advantages the growth of the invader such that it takes over more rapidly. This is contrary to the behaviour of the conditional MFPT at very high fitness ($w > 10$): the average conditional MFPT of invasion increases with fitness difference increases, see Section F in S1 Appendix for details.

## Discussion

Spatial exclusion can significantly alter the competitive dynamics between species in densely populated bacterial communities. To understand bacterial competition in confined geometries, we investigated the competition between two cellular species confined to an open 1D microchannel lane. To this end, we have developed a spatial exclusion model that explicitly

accounts for the mechanical exclusion between cells, in contrast to non-spatial well-mixed models like the paradigmatic Moran model.

In this study, we examine two competition scenarios of populations of two species in a one-dimensional channel. In the first case, each species population is segregated to separate sides of the channel, resulting in a single boundary between the two populations. The MFPT analyzed in this scenario corresponds to the time for the system to fixate; in other words, the fixation is not conditioned on the fixation of either species.

In the one boundary scenario, we find that the probability of species fixation in the spatial exclusion model shows a much sharper sigmoidal dependence on the initial fractional abundance in contrast to the Moran model where the corresponding probability is equal to its initial fractional abundance. The inflection point of the sigmoidal curve—where the probabilities of fixation of either species are equal—is located at the initial abundance $f = 1/2$ for neutral populations without relative fitness difference ($w = 1$), but shifts to lower initial abundances of the fitter species ($f_{eq} = 1/(1 + \sqrt{w})$) at higher values of the relative fitness difference. This result in the spatial model is different from the equivalent quantity in the Moran model: the equiprobable takeover abundance in well-mixed models depends on the relative fitness difference as well as the system size $N$, see Section C in S1 Appendix. With an increase in the population size, $N$, the sigmoidal curve approaches a step function, effectively setting a threshold in the initial abundance above (below) which the cells will always (never) fixate.

The overall dynamics of competition can be characterized by the mean time of fixation (unconditioned on either species dominating the channel). The mean fixation times in the spatial exclusion model are sped up in comparison to the predictions of the Moran model. The fixation time is essentially independent of the total population size for most initial abundances compared to the well-mixed models (where the MFPT scales linearly with system size), and is dominated by a quasi-deterministic exponential escape towards fixation arising from skewed collective growth of the cells toward channel exits. The fixation time shows a weak dependence on population size that is well approximated by an asymptotic logarithmic scaling in population size only for the initial abundances near the inflection point of the fixation probability curve, where the maximum timescales are observed (see Figs 2C and 3C).

Although the species with a relative fitness difference maintains a competitive advantage within our spatial exclusion model, this competitive advantage does not affect the fixation dynamics to the same extent as it does within the Moran model. Indeed, within the spatial exclusion model, the equiprobable takeover abundance occurs at lower initial fractional abundances of the fitter species than in the neutral model, see Fig 3A. However, the competitive advantage a bacterial species gains by having higher relative fitness than its competitor is significantly less in this constrained environment than in a well-mixed system devoid of spatial limitations. Although births of the more fit species happen more frequently, the spatial organization of the cells make their deaths more frequent as well (as they fall out of the channel) resulting in a reduction in the competitive advantage compared to the well-mixed model.

For the 1D channel with fitness differences, we showed that one species is deterministically favoured to out-compete the other when the initial fractional abundances are not close to the equiprobable takeover abundance. By contrast, for initial abundances in a region close to the equiprobable takeover abundance, the dynamics are dominated by stochastic diffusion. Either species is roughly equal likely to take control of the channel within this region with the two opposing species competing in a "tug-of-war". We find that the "tug-of-war" concludes with one species outcompeting its rival in times that scale approximately as $\mathcal{O}(\log N)$ when starting in the diffusive regime. This is significantly different from the paradigmatic Moran model which predicts a $\mathcal{O}(N)$ fixation.

More generally, we have developed a heuristic framework to approximate the asympotic dependence (at large $N$) of the mean first-passage times in a "tug-of-war" process with concave potentials. More precise approaches to this framework will be considered in the future.

The heuristic framework for calculating the asymptotic MFPTs is of general interest and has applications well beyond bacterial population dynamics. The results of the competition between the species are due to these "tug-of-war" dynamics engendered by the skewed collective growth, which may be applied to other systems with drift and diffusion terms leading to a concave effective potential as in Fig 4A. For instance, the transport of organelles and other cellular cargo has been described by a "tug-of-war" wherein competing sets of molecular motors pull in opposite directions with the drift depending on the number of motors on either side of the cargo, much like our model [92]. Moreover, competition between populations of cancerous and healthy cells display a "tug-of-war" effective potential that recovers a probability of cancer development that is sinusoidal as a function of the initial fractional abundance of cancerous cells, as in the spatial exclusion model [93]. Our methodology can predict the probabilities of and mean times to a clinical outcome determining the rapidity of the disease.

As expected, an increase in the relative fitness increases the probability of fixation and shortens the fixation time of the species with the higher fitness. However, mechanisms that convey selective advantage in division rates can be difficult to maintain due to their strain on metabolism and increased resource costs [94, 95]. As shown in Section D in S1 Appendix, the decrease in the fixation times of a higher fitness species conditioned on successful fixation can be quite small compared to the neutral case. In the long term, this introduces an evolutionary tradeoff between the increased fitness and the cost of maintaining it.

In this paper, we used a minimalistic representation of the skewed collective growth reflecting crowding within the channel. Different models with higher or lower degrees of bias are possible, including models where the basal division rate directly depends on the local conditions such as adhesion to the walls, and the position of the cell in the channel especially in long channels [67, 96]. These models will be studied in the future.

In the second scenario studied in this paper, an invading individual is inserted into a homogeneous inhabitant population in the channel at any location, creating two inter-species boundaries at either side of the invader. This scenario corresponds to the dynamics of the two-boundary invasion model wherein we explore the probability and time for the invader to successfully fixate in the channel. If the invasion is on the edge of the channel, this is identical to a special case of the single boundary case. For these invasions, we calculated the probability and the MFPT of a successful invasion for different invasion locations, assuming that the invasion is likely to occur at every point in the channel with equal probability. Overall, an invader with a relative fitness difference increases the likelihood that an invasion at any location is successful, with the invasions most likely to succeed occurring in the middle of the channel. Similar to the Moran model, the probability of a successful invasion (averaged over all initial conditions in the channel) decreases as the total population size increases.

*Koldaeva et al.* investigated the loss of lineage diversity in a channel for a neutral model. In a single lane, their results describe the mean first-passage time in which the progeny of a bacterium takes over the whole lane. This is closely related to the invasion scenario studied in our paper, where we calculate the first-passage time for an invading bacterium to fixate starting from different initial locations in the channel. Notably, for the neutral case, the mean time of successful neutral invasion when averaged across all possible initial locations in the channel weighted with the invasion probability from that location is numerically very close to the mean time in which diversity is lost in *Koldaeva et al.*'s study. Both their findings and our results in Fig 5D agree that this aggregate diversity loss time grows approximately logarithmically with system size.

As discussed above, the time for a species to fixate in the one-dimensional boundary (two domain) case also exhibits logarithmic scaling with system size at the equiprobable takeover abundance, as shown in Fig 2C. There is a connection between these two timescales, accounting for their similar scaling behavior. Given that most successful invasions occur when initiated from the center of the lane, as depicted in Fig 5B, the average invasion time (Fig 5D) is dominated by the invaders originating near the center. Alongside this, the fixation time of an invader starting near the center is the same (up to a factor) as the fixation time in the one-boundary case starting at the location of equiprobable abundance in the neutral case. Thus, the logarithmic scaling of the average invasion time is closely linked to the logarithmic scaling of the fixation time in the one-boundary case.

In recent years, microfluidic monolayer devices (MMD)—such as mother machines, chemostats, etc.—have been designed to study single cell bacterial growth and generational dynamics [67, 96, 97]. Our results model the behaviour of competing populations in a single-lane, open chemostat and could be directly tested within such a device. However, our spatial model can also be extended to more complex 2D MMDs that support multi-lane channels. Pill-shaped bacteria, such as *E. coli*, are observed to grow constrained to 1D lanes within wide, open-ended microchannels [57, 58, 98]. As a first approximation, the larger channels can be viewed as many 1D lanes that interact through rare immigration events. In this simplified view, the dynamics of channel fixation can be decoupled from the lane fixation if the time between lane invasions is longer than the fixation time within a lane. An interesting direction for future work would be to combine the probability of a successful invasion found in Fig 5 with a rate of invasion to model 2D competition and check if the logarithmic scaling previously reported is recovered as we extend the model to higher dimensions.

In summary, we have shown that explicitly incorporating spatial interactions arising from cell growth and division within dense bacterial populations can have important consequences for both the overall composition and the rate of species exclusion from the system. In densely packed microenvironments, bacteria will align into lanes to maintain their growth in the confined geometries of the space they inhabit. In the one-dimensional geometry of a channel, the skewed collective growth of cells fosters the "tug-of-war" dynamics between two populations, showcasing a distinct form of competition compared to well-mixed populations. Our results describe the outcome of the competition for space between species, predicting the timescale in which the competition terminates and the likelihood that a species survives in the ecosystem. Our results provide insights into the processes involved in the formation and maintenance of complex bacterial ecosystems such as biofilms, intestinal flora, or various persistent infections. Likewise, the mathematical techniques developed here may more broadly be applied to a range of competitive dynamical systems, from cellular transport to cancer.

## Supporting information

**S1 Appendix. Section A. Calculation of the discrete mean first-passage time to fixation**. We review the calculation of the probability and exact mean first-passage time for the discrete models discussed in this main text. **Section B. The Fokker-Planck approximation**. The Fokker-Planck approximations corresponding to the continuous limit of the exact discrete dynamics are derived. **Section C. The Moran model**. Details of the classical Moran model with fitness differences are summarized. The differences between results of the Fokker-Planck equation and the master equation are explored. **Section D. The spatial exclusion model**. Further details of the spatial exclusion model are clarified, including the MFPT of fixation conditioned on the success of one of the two species. Intermediate fitness differences are plotted. A comparison of the Fokker-Planck and master equation results is shown. **Section E. Average**

**probabilities and MFPT**. The MFPT and probability of fixation are averaged over all initial fractional abundances for the one-boundary problem. **Section F. Invasion into a fixated channel**. The mathematical details of the inva.sion dynamics discussed above are outlined. Additional results on a minimal invasion from the side of the channel are also shown. **Section G. The heuristic asymptotic MFPT approximation**. The asymptotic in $N$ behaviour of the maximal MFPT at the equiprobable takeover abundance is explored. A mathematical derivation involving an approximation to the MFPT of fixation is investigated.
(PDF)

## Acknowledgments

The authors are indebted to the members of the Milstein, Zilman and Goyal labs for numerous discussions.

## Author Contributions

**Conceptualization:** Jeremy Rothschild, Tianyi Ma, Joshua N. Milstein, Anton Zilman.

**Formal analysis:** Jeremy Rothschild.

**Funding acquisition:** Joshua N. Milstein, Anton Zilman.

**Investigation:** Jeremy Rothschild, Tianyi Ma, Joshua N. Milstein, Anton Zilman.

**Methodology:** Jeremy Rothschild, Anton Zilman.

**Project administration:** Joshua N. Milstein, Anton Zilman.

**Software:** Jeremy Rothschild.

**Supervision:** Joshua N. Milstein, Anton Zilman.

**Validation:** Joshua N. Milstein, Anton Zilman.

**Visualization:** Jeremy Rothschild, Tianyi Ma.

**Writing – original draft:** Jeremy Rothschild, Joshua N. Milstein, Anton Zilman.

**Writing – review & editing:** Jeremy Rothschild, Joshua N. Milstein, Anton Zilman.

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
