## [Decision Letter · Decision Letter 0]

16 Mar 2023

Dear Dr. Zilman,

Thank you very much for submitting your manuscript "Spatial exclusion leads to tug-of-war ecological dynamics between competing species within microchannels" for consideration at PLOS Computational Biology.

As with all papers reviewed by the journal, your manuscript was reviewed by members of the editorial board and by several independent reviewers. In light of the reviews (below this email), we would like to invite the resubmission of a significantly-revised version that takes into account the reviewers' comments.

While the reviewers felt that the manuscript had interesting results, with Reviewer 1 in particular expressing enthusiasm, both reviewers raised very substantial concerns about the content. Reviewer 1 pointed out that throughout the manuscript, the interpretation of fixation times is confused, and that all the simulations are for extremely strong selection. Reviewer 2 pointed out that the manuscript does not make clear its close connection to the existing theoretical literature, and that it is unclear whether the results represent a sufficient enough advance on that literature to warrant publication. Addressing these points, as well as the other important points raised by the reviewers, would require extensive rewriting of the manuscript and additional analysis, such that the resulting manuscript would be quite different from the current version.

We cannot make any decision about publication until we have seen the revised manuscript and your response to the reviewers' comments. Your revised manuscript is also likely to be sent to reviewers for further evaluation.

Sincerely,

Daniel Weissman, Ph.D.

Guest Editor

PLOS Computational Biology

James O'Dwyer

Section Editor

PLOS Computational Biology

While the reviewers felt that the manuscript had interesting results, with Reviewer 1 in particular expressing enthusiasm, both reviewers raised very substantial concerns about the content. Reviewer 1 pointed out that throughout the manuscript, the interpretation of fixation times is confused, and that all the simulations are for extremely strong selection. Reviewer 2 pointed out that the manuscript does not make clear its close connection to the existing theoretical literature, and that it is unclear whether the results represent a sufficient enough advance on that literature to warrant publication. Addressing these points, as well as the other important points raised by the reviewers, would require extensive rewriting of the manuscript and additional analysis, such that the resulting manuscript would be quite different from the current version.

Reviewer's Responses to Questions

**Comments to the Authors:**

Reviewer #1: This manuscript is about evolution in a spatial environment, and comparing the resulting results with a classic well mixed Moran model. I really liked it. It is a neat analysis of a realistic 1-dimensional model, which results are interestingly very different from the well-mixed case. However, I feel there are several aspects that should be improved. In particular, the scope of the model could be more explicit in the abstract and authors summary. More importantly, there is a vocabulary confusion between the fixation time (of either types), or the fixation time conditioned on the mutant success, which leads to some inadequate interpretations. There are some other confusing notations. There are also some statements that are not well supported by the results, and some equations that have issues (but that may be due to some typos). Below I detail each point, and finish by some smaller comments.

*In the abstract, there should be some mention that there is no death, but loss via the open ends of the channel. For instance, complement the sentence ending by "within a one-dimensional open microchannel." with something like ", in which cell loss happens only at the channel ends.". In the author summary, there is no mention that this is a 1-dimensional result, which I think is important.

*There is a vocabulary confusion between the fixation time (of either types), and the fixation time conditioned on the mutant success. The MFPT is defined as the time of absorption in a state with either only all WT or all mutant. Most figures are showing this MFPT; only a few figures in appendix show the fixation time conditioned on mutant success. Yet, it seems that most commentaries are confusing these two different quantities.

- In the abstract for instance, the "the fixation times increase as the relative fitness differences between species increase." gives the impression that the fixation of a mutant goes slower with a higher fitness, whereas actually this is the time for fixation of either the WT or the mutant, and as the probability for the mutant to fix increases, the time to fixation for either types increases, because the time for fixation of the mutant is longer than the time of fixation of the WT. There are only 2 figures, panels 3F and 6F in appendix, that show, both in case when the mutants are on one end of the channel, there is indeed a very weak increase with fitness of the fixation time conditioned on success; but in a context in which the fixation probability increases greatly with fitness, so that actually in the few cases in which the mutant fixes at low fitness, the process has been very lucky, and thus quick. But my supposition is that for the case of a mutant arriving anywhere in the channel, by migration from a neighboring channel, the conclusion would be reversed. In this case, the successful invasions are from the middle of the channel, and the larger the fitness, the faster they would occur (but that would need to be checked).

- In the abstract also: "Surprisingly, in the spatial exclusion model, successful fixations occur on average more rapidly in longer channels": actually in the whole manuscript only unconditional fixation time is shown, except for panels 3E, 3F, 6E and 6F in appendix, and for 6E and 6F, they actually show that in the model considered in these figures, the fixation time conditioned on mutant success does increase with N.

- from line 355: "Additionally, we find that the average MFPT of a successful invasion in the spatial model increases with fitness advantage (...) This is contrary to the intuition that higher selective advantage should accelerate the fixation of the invasive species because the fitter invader grows more rapidly.(...)" Actually what increases with the fitness is the average unconditional MFPT, thus actually the main effect is likely that at higher fitness, the mutant is more likely to fix, and fixation of a mutant takes longer than loss of the mutant.

- section titled "Invasions are less likely to succeed due to spatial exclusion, but succeed on shorter timescales" whereas almost all results are on unconditional MFPT, and the only cases where MFPT conditioned on invasion success is shown, sup figures 3EF 6EF, there are actually in 6EF parameters for which the success MFPT is longer in the spatial model than in the well mixed model.

- from line 447 "Furthermore, competition between species with higher fitness differences show longer fixation times despite the fact that one of the species has a greater likelihood of dominating the channel. The slower

fixations are owing to slower diffusion relative to drift around the point of equiprobable abundance." As said before, not proved. "This means that not only must a species preserve a potentially more resource intensive strategy to gain an advantage, but it must sustain this strategy for longer to insure it dominates. Thus, the overall competitive advantage provided by an increase in fitness must be weighed against these two competing effects." Given the discussion is on the unconditional MFPT, not the MFPT conditioned on success, the last part does not make sense.

- around line 461: "the mean invasion times averaged over the initial channel location decrease with system size. These fixation times accelerate in longer channel because successful invasions with larger population sizes are reliant on the the invader species remaining near the center of the channel, which fixates more rapidly. Consequently, successful invasions in longer channels happen less frequently, but those that are successful are more rapid on average." There is no figure with MFPT conditioned on success with the model with invasion anywhere in the channel. So no basis for making such a claim.

*Other notations that are confusing:

- Usually s is used for the increase in fitness, not for the fitness itself (usually the fitness is noted as w=1+s). It is defined clearly in the manuscript, so it's ok, but it is a bit unusual notation.

- the "abundance at which both species are equally likely to take over the channel" is called "equiprobable abundance", which is confusing if the reader has missed the definition. It's a bit longer to write, but "equiprobable takeover abundance" would be clearer.

- sometimes x represents the frequency of mutants, sometimes it represents the position in the channel. It would be much clearer to use one notation for frequency, and another for space, consistently in the whole manuscript.

*There are a few things that should really be corrected, either statements that are not supported by the data, or equations (possibly typos):

- line 295 "In other words, the MFPT away from the equiprobable abundance will asymptotically decrease with fitness differences." Figure 3B: this is ok on the right side of xmax, but on the left side of xmax, it's actually the reverse, at a given x, the MFPT increases with fitness differences.

- equation (12): to be equal to the right hand side, the left hand side should be -N exp(-U(x)), not -1.

- equation (16): the last 2 terms are the same, one of them is probably a typo.

- equation (9) of the sup: N instead of n twice.

- just after this equation "p(n,t|n0,t+delta_t)=p(N,t-delta_t|n+1,0)" does not make sense. I suppose "p(n,t|n0,delta_t)=p(n,t-delta_t|n0,0)" is what is meant.

- equation (18) sup, I tried to redo, I think it's N^2, and not N.

*Smaller comments:

- s values are huge! A 10 or 100-fold increase of fitness is not frequent, except in cases with a drug is applied and the mutation confers drug resistance (and in which cases, realistically, there is often bactericide effects on top of the bacteriostatic effects)... In appendix, there is a smaller change of fitness, 3-fold, that is shown, but it is still really large. Maybe it would be good to also show the results (possibly in appendix) for more moderate (and more likely to be observed!) changes in fitness, let's say for instance 1.1.

-line 338 "To provide a global measure of invasion success, we calculate the average invasion probability and the average MFPT to invasion, averaged over all possible initial locations of the invasion event. Surprisingly, this average invasion probability in the neutral spatial model is identical to the invasion probability of the neutral Moran model (s = 1) (i.e., it is equal to 1/N as shown in Fig 4C)." Actually I don't think it's surprising at all. In this model, the neutral invader can arise equally at each position in the channel. One of the N cells of the channels will be the ancestor of the whole population in the long term. Thus the invasion probability averaged on the initial position is 1/N.

- line 434: "the fixation time show a weak dependence on population size that is well approximated by an asymptotic logarithmic scaling in population size." : A logarithmic scaling in population size is plausible, but it's not fitted, and on such a small range (maximum 100 fold difference in N), it can be difficult to tease apart different possibilities.

- page 17 in appendix, the paragraph commenting on fig 6C&E, the invasion probability for the Moran process being independent on N is not true for s=1.

*tiny comments

- fig 1C: the schematic is correct, but putting "death right" on the left side, and "death left" on the right side is kind of confusing on a first read.

- legend of fig 3B should give the reference of the equation for the dark dashed line.

- in the discussion in my opinion the initial summary of the findings should give a break down of the different models studied: case of a population with mutants on one side, WT on the other side; case of a mutant arising, either from an open end, or from anywhere with equal probability along the channel;

- The first part of the appendix (pages 1 to 3) does not seem to relate directly to the rest of the manuscript.

- it would be good if references in the main text to the supplementary were more precise, towards a part of the supplementary, rather than to the whole thing.

- fig 3A sup: the legend for the colors is put at a position where s is not visible.

- after 7.1, somewhere there is "thee" instead of "the"

Reviewer #2: Manuscript "Spatial exclusion leads to tug-of-war ecological dynamics between competing species within microchannels" by Rotschild et al. studies a spatially explicit population model in which individuals (cells) are arranged into a line and push each other when reproducing, leading to expulsion of cells at the two open boundaries. The authors study fixation probabilities and mean fixation times in different scenarios, including neutral and non-neutral competition.

I found that this study contains some interesting results. On the other hand, I have some major concerns, that cast doubts on whether the manuscript is appropriate for Plos Computational Biology. As elaborated below, the authors completely ignored the existing theoretical literature on population dynamics with pushing dynamics. Indeed, some of the results presented here are known in the literature, which somehow limits the novelty of this work. Moreover, the manuscript is presented in a rather technical style which does not seem appropriate for PCB.

Major issues:

(1) The abstract and introduction give the impression the authors are the first to theoretically study population dynamics incorporating pushing dynamics. But this is not the case. An early, important study in the context of cancer progression is Nowak, Michor, Iwasa, PNAS (2003), followed by Allen and Nowak, JTB (2012). It is more troubling that, in the introduction, the authors mention Koldaeva et al. (Ref. [66]) just as an experimental motivation for their study. In fact, Ref. [66] includes a systematic theoretical study of the neutral case - including exact expressions for the fixation probability and the logarithmic dependence of the fixation time on the system size. The authors should properly refer to the relevant literature and clarify what is new here.

(2) I am quite familiar with the mathematical techniques used in this work and therefore could follow its logic. However, I feel that the authors made little effort to make the manuscript accessible to a broader audience. In particular, most of the results are presented in a rather technical fashion, without clearly explaining their biological significance. I feel that this is not the standard for PCB and I therefore recommend a substantial revision, having in mind readers that are less familiar with the math. One approach could be to separate the main messages of the paper from the mathematical derivations and analysis.

Other issues:

- The Methods section begins with a lengthy review of the Moran model, which seems unnecessary since the model is so well established. This part can be drastically shortened. In contrast, the model with two boundaries used for Fig. 4 should be at least briefly defined in this section (although I agree with the choice of leaving the mathematical details for the SI).

- Line 234, "grow only sub-polynomially". The fixation time is actually logarithmic, as exactly proven in Ref. [66]. It is surprising that the logarithmic scaling is mentioned in the abstract, but in the results this logarithmic dependence is not really fleshed out.

- Line 334-337 The fact that mutations in the center of the channel are much more likely to reach fixation is, again, exactly proven in Ref. [66], although only in the neutral case.

**Have the authors made all data and (if applicable) computational code underlying the findings in their manuscript fully available?**

Reviewer #1: Yes

Reviewer #2: Yes

PLOS authors have the option to publish the peer review history of their article (what does this mean?). If published, this will include your full peer review and any attached files.

Reviewer #1: No

Reviewer #2: No
---

## [Decision Letter · Decision Letter 1]

17 Jul 2023

Dear Dr. Zilman,

Thank you very much for submitting your manuscript "Spatial exclusion leads to tug-of-war ecological dynamics between competing species within microchannels" for consideration at PLOS Computational Biology. As with all papers reviewed by the journal, your manuscript was reviewed by members of the editorial board and by several independent reviewers. The reviewers appreciated the attention to an important topic. Based on the reviews, we are likely to accept this manuscript for publication, providing that you modify the manuscript according to the review recommendations.

Both reviewers agree that the manuscript is greatly improved from the previous version and are optimistic about it being able to be suitable for publication. They also both agree that the authors should do more to emphasize the continuity between the results of Koldaeva et al and those in the present manuscript. Reviewer 2 additionally points out that the responses to the two main points of their previous review are insufficient. First, small fitness differences are only treated briefly in the supplement, without the results being integrated into the main text. For instance, in the main text it states that (line 278) "higher fitness differences in the Moran model reduce the MFPT unconditioned on the success of a species", but this is not true for small fitness differences, as shown in the supplement. Second, several references to fixation times are still unclear.

Trying to synthesize all the reviews and my own impressions, I think that the manuscript needs to spend less time on trying to draw contrasts with other models and more on helping the reader gain an intuitive understanding of the model and its conclusions. The goal should be explain why every result is natural and makes sense. If the authors find any results to be surprising or counter-intuitive, they should try to see if there's an alternative framing that makes them intuitive. A few specific suggestions to get started:

1. The key ingredient here is the biased pushing or whatever you want to call it. This creates strong spatial variation in reproductive value (Fig 4B, s=1). Cells in the middle get to reproduce for many generations before they have any chance at all of losing any indviduals, while cells on the edge are doomed. This manuscript deals with the consequences of this spatial variation for evolution. This needs to be explained as early as possible---if it's too complicated for the abstract, at least in the Introduction, and then with a simple intuitive description in the model section. And it should be included in Fig 1. (I would suggest replacing Fig 1 entirely with just an illustration of the collective pushing effect.) This should allow all the subsequent results to be explained fairly intuitively. (For instance, because the pushing induces an effective strong selection against cells at the edge, very strong selection is needed to allows cells to invade from the edge.)

2. When discussing MFPTs, there always needs to be a clear distinction between changes driven in by changes in dynamical time scales and those driven by changes in fixation probabilities. This is true even in the second part where the manuscript is conditioning on invader fixation, because it is still averaging over different initial locations. For instance, in the neutral results shown in Fig 4D, I'm guessing that the dynamical time scales (i.e., the average fixation times for lineage originating at specific locations) are actually increasing with N or remaining constant, and the decrease in MFPT is because lineages in disfavored starting locations, whose fixation would be slower, have lower fixation probabilities as N increases. If this is true, I think that the presentation of MFPT here is going to tend to lead readers astray. Even if it's not true, the more important point is that the text should make it totally clear what's going under the hood. A similar comment applies for the effect of increasing invader fitness in the same panel.

So overall, I think that only very minor revisions are needed to the results, but major revisions are needed to the presentation.

One very minor additional comment: throughout, "fixate"/"fixated" should be replaced by "fix"/"fixed". "Fixation" is the only "fixat*" term that is usually used.

Sincerely,

Daniel Weissman, Ph.D.

Guest Editor

PLOS Computational Biology

James O'Dwyer

Section Editor

PLOS Computational Biology

Both reviewers agree that the manuscript is greatly improved from the previous version and are optimistic about it being able to be suitable for publication. They also both agree that the authors should do more to emphasize the continuity between the results of Koldaeva et al and those in the present manuscript. Reviewer 2 additionally points out that the responses to the two main points of their previous review are insufficient. First, small fitness differences are only treated briefly in the supplement, without the results being integrated into the main text. For instance, in the main text it states that (line 278) "higher fitness differences in the Moran model reduce the MFPT unconditioned on the success of a species", but this is not true for small fitness differences, as shown in the supplement. Second, several references to fixation times are still unclear.

Trying to synthesize all the reviews and my own impressions, I think that the manuscript needs to spend less time on trying to draw contrasts with other models and more on helping the reader gain an intuitive understanding of the model and its conclusions. The goal should be explain why every result is natural and makes sense. If the authors find any results to be surprising or counter-intuitive, they should try to see if there's an alternative framing that makes them intuitive. A few specific suggestions to get started:

1. The key ingredient here is the biased pushing or whatever you want to call it. This creates strong spatial variation in reproductive value (Fig 4B, s=1). Cells in the middle get to reproduce for many generations before they have any chance at all of losing any indviduals, while cells on the edge are doomed. This manuscript deals with the consequences of this spatial variation for evolution. This needs to be explained as early as possible---if it's too complicated for the abstract, at least in the Introduction, and then with a simple intuitive description in the model section. And it should be included in Fig 1. (I would suggest replacing Fig 1 entirely with just an illustration of the collective pushing effect.) This should allow all the subsequent results to be explained fairly intuitively. (For instance, because the pushing induces an effective strong selection against cells at the edge, very strong selection is needed to allows cells to invade from the edge.)

2. When discussing MFPTs, there always needs to be a clear distinction between changes driven in by changes in dynamical time scales and those driven by changes in fixation probabilities. This is true even in the second part where the manuscript is conditioning on invader fixation, because it is still averaging over different initial locations. For instance, in the neutral results shown in Fig 4D, I'm guessing that the dynamical time scales (i.e., the average fixation times for lineage originating at specific locations) are actually increasing with N or remaining constant, and the decrease in MFPT is because lineages in disfavored starting locations, whose fixation would be slower, have lower fixation probabilities as N increases. If this is true, I think that the presentation of MFPT here is going to tend to lead readers astray. Even if it's not true, the more important point is that the text should make it totally clear what's going under the hood. A similar comment applies for the effect of increasing invader fitness in the same panel.

So overall, I think that only very minor revisions are needed to the results, but major revisions are needed to the presentation.

One very minor additional comment: throughout, "fixate"/"fixated" should be replaced by "fix"/"fixed". "Fixation" is the only "fixat*" term that is usually used.

Reviewer's Responses to Questions

**Comments to the Authors:**

Reviewer #1: I have a bit mixed feelings on this revision. I think overall that the manuscript is interesting, and could be worth publishing after revisions. And improvements were made. However, I feel that the authors did not fully take into account the comments that were made and were sometimes a bit careless. I usually do not spend as much time on a revision. My comments are in 4 main categories: the relation with the Koldaeva article; the question of the fitness; the clarity; and some other points. In all the references below, line numbers are the line numbers of the document with the highlighted changes; and the numbers of the comments of the reviewers are the numbers used by the authors.

* The relation in comparison to previous study by Koldaeva et al could be clarified. I am grateful to the other reviewer for looking at this article more carefully, I had not realized they had already quite important analytical results in this other article, on top of the experimental data. I agree however with the authors that this manuscript still brings something new, in particular another observable and extending to the case of mutants with a selective advantage.

- in the answer to the comment 1 of the other reviewer, the authors state that "Koldaeva et al consider a channel with two openings, akin to ours, but focus on a rather different quantity – average time to fixation of the lineage of one of the bacterial cells initially present in a one-dimensional lane. Within a number of approximations employed in the paper the dependence of this quantity on N emerges to be approximately logarithmic. However, we stress that this is an entirely different quantity compared to ours – fixation of one of the two species present initially regardless of progeny of how many individual cells remain present. The fact that it’s also logarithmic is a appears to be coincidental." I don't find the wording very clear. Koldaeva et al consider the diversity at a given time, and the time it takes so that one initial bacteria is the ancestor of all the bacteria in the system; they also calculate the invasion probability of one bacteria at a given position in the neutral case. The authors in this manuscript look at the probability for one bacterial type to take over, in the case with only 2 bacteria types. They also calculate invasion probabilities for one bacteria at a given position, but go beyond the neutral case.

- I disagree that the logarithmic dependence is completely coincidental. Here they show that the time for fixation at the initial frequency which has the longest average time to fixation (the half/half for the neutral case) depends logarithmically on N. For the quantity looked at by Koldaeva, the time to fixation of a lineage from one of the N initially present bacteria, in most cases the bacteria which lineage fix comes from the middle of the channel, and thus the typical time it takes for this lineage to fix on one side will be the same ordre of magnitude than for it to fix on both sides. So, at least heuristically, it makes sense that both quantities have the same overall dependence on N.

- A point that I had not noticed in my first review, due to my insufficient reading of the Koldaeva reference. It concerns that (starting from line 203) "in open-ended micro-channels, it has been experimentally observed that cells are more likely to grow in the direction of the closer channel opening because fewer cells need to be pushed in that direction. The probability for a cell to grow towards one of the openings was observed to scale linearly with the number of individuals between the cell location

and the other opening [57]" (57 is the Koldaeva reference). In this manuscript, pleft=(i-1)/(N-1), whereas in Koldaeva, pleft=im/(N-1)+(m(N+1)+N-1)/(2(N-1)), with m a parameter. If m=0, then there is no bias towards the end of the channels. If m=1, pleft=(N-i)/(N-1), which is, with some confusion on the meaning of left/right, the same than this manuscript. However, the fit in Koldaeva data gives m=0.6. The authors should state explicitly that they have chosen a limit of the theoretical analysis of Koldaeva, which is for somewhat larger impact of the numbers of individuals on each side than supported by the data.

- Paragraph lines 81-87. "Using heuristic analysis, the study showed that the diversity of the genetic lineage can quickly be lost in the channels where identical cells grow in lanes. However, a full mechanistic description of the dynamics of two species populations with fitness differences competing due to spatial exclusion in this one-dimensional space has not been elucidated." I don't think this is very fair to the Koldaeva study. They use some heuristics, but they also make some analytical calculations. And their model is also "fully mechanistic", though one very valid point is that they are not considering fitness differences, contrary to this manuscript.

- The authors could compare their solution (both probability and time to fixation) to the Koldaeva result for the invasion case of a neutral bacteria.

* The study of cases with more reasonable fitness differences is a bit weak. They add one figure with less enormous fitness differences for the first question, but nothing for the invasion process (neutral, 10 fold, 100 fold more fit, which is enormous). This question of the fitness differences was stressed by the editor, and in my mind, the authors did really a minimal job on the subject.

Besides, I am annoyed by the authors not answering properly to my comment 8. It was not a crucial comment, I was just commenting than in general, s is the notation not for the fitness itself, but for the difference in fitness: if one species has fitness "1", and the other one has a fitness of 1+s; whereas in this manuscript the reproduction rate is 1 for one species, and s for the other. The authors replied saying that they are using a convention used by other authors. They refer to 2 sources. In the PRE article they cite, actually when s is introduced, "In this spirit, we assume that the relative fitness of alleles X on island i is [WX ]i = 1 + s ρi, and correspondingly [WY ]i = 1 + s σi for allele Y on island i, where the strength of selection parameter, s, is small. ", which is actually exactly what I was suggesting, and *not* the choice of the authors. They also cite a book. And in this book, I could find the notation s used in 3 examples. In one, it has nothing to do with fitness, s1 and s2 are some frequencies. In another one, s is indeed a fitness, as used in this manuscript by the authors, but I think that in that example s was chosen because it is the relative fitness of a superinfection. And there is a 3rd example where s is used, for a fitness 1 + s x, thus more in line with the usual notation of s as the relative increase in fitness. Notations are not that important, but it may be clearer for the readers to use more common conventions. I found the authors'answer unpleasant, because this gives the impression that either they did not care and put whatever references they had in mind without checking; or they knew their references were not particularly supporting to their choice of notation and thought I would not check.

* Though it is really quite improved compared to previous version, I think there are still some places where some aspects are unclear.

- In the abstract: "In non-neutral cases, (...) and the maximal system fixation times increase as the relative fitness differences between species increase." is confusing for 2 reasons: it does not say that this fixation time is not conditioned on the success of one particular species, and it does not say maximal relative to what (it is the maximal unconditional fixation time, relative to the location of the initial boundary between species). I'm not sure the result is so fundamental to be stated in the abstract, given the space it takes to state it properly.

- I reiterate that I think that stating that the results are in 1-dimension is important enough to be mentioned in the authors' summary, and not just the abstract. The sentence "In this study, we theoretically and computationally study the competitive dynamics of two bacterial populations competing for space in confined environments" could be transformed to something like "In this study, we theoretically and computationally study the competitive dynamics of two bacterial populations competing for space in a dimensional confined environment".

- paragraph lines 525-538. The discussion there is framed as if there were some trade-off between fitness (replication rate) at a given time and "strain on metabolism" and "resource costs" (that would imply slower replication at a later time?). I am unsure of what relevant examples would be for bacteria. Moreover, it is framed around the increase with fitness of the time to fixation (of either species) at the equiprobable takeover initial abundance; however, the time to fixation conditioned on success, at a given initial abundance, is always smaller for a higher fitness. I don't understand the message that this paragraph is supposed to convey. In their answer to my point they numbered 6, the authors state that "What’s more, the unconditional time to fixate at the equiprobable takeover abundance increases with fitness difference. At this abundance, the fitter species has the same probability to successfully takeover as the less fit species, yet the competition now lasts longer than at lower fitness differences. The populations must make a decision on the strategy to undertake based on the trade-off between higher success in the chamber and shorter competition times." I really disagree on this. Let's suppose that the population is able to make a decision (which can be doubtful for bacteria). At a given initial relative abundance of one species, then an increase in fitness does increase the chance of success and does reduce the time to fixation conditioned on success. It is only looking at the initial abundance for equiprobable takeover, that shifts with fitness, that the conclusion is reversed.

- Around line 304-311 of the document with highlighted changes: "Counter-intuitively, in the spatial exclusion model, although a greater fitness is generally associated with a higher likelihood of population fixation, increased fitness does not lead to faster fixations.". I think it would be clearer to use all the results and say something like: "In the spatial exclusion model, though a greater fitness always leads to faster fixation conditioned on the species success (see fig 4B of the supplementary material), this is not always the case when considering the unconditional fixation probability."

- section "Quantitative investigation of different dynamical regimes in large systems competing in a tug-of-war": is it back to the case where one side is one species, the other the other species? Or are we still in the invasion problem as in the section just before? If it's the one boundary system, it would be much clearer for the reader to say explicitly at the beginning of this section that it is now back to the first set-up. If it's the invasion set-up like previous section, then many things don't make sense.

- Overall in this manuscript, the level of detail of the calculations is ok, but I found it very hard to follow the calculations of the supplementary part 7.

* Other comments (random order).

- their reply to my comment 15: they did make a change, but it is still not ok. It is not true that the transformation is (neither should be!) p(n, t|n0 , t + ∆t) = p(n, t − ∆t|n0 , 0), it's p(n, t|n0 , ∆t) = p(n, t − ∆t|n0 , 0).

- Figure 4 of the sup: how many simulations for calculating the conditional fixation time? Any indication of the standard deviation between first passage time? (I ask because when the low fitness species starts at a low frequency, its probability of fixation is very small).

- frankly I don't think that part 5 of the appendix is interesting: averaging on the initial frequencies, with uniform probability. Why this choice? This gives some aggregate number, but I don't think it has any physical meaning.

- line 307: "location dependent" -> it's not the location, but the initial frequency f. That frequency translates in a location for the interface, as here it is the case when each species is on one side, but location is more relevant for the invasion case.

- Equation just after line 324 should be with f instead of x

- line 422 "a regime relevant for many experimental systems that are often comprised of large numbers of individuals" Could you be more explicit about such systems? Microfluidic channels are usually not very long, mostly for nutrients to diffuse enough.

- Figure 5b. The legend directly on the figure does not match the change in variable names.

- The discussion paragraph between lines 572 to 580. First line 573 it mentions invasion, then line 576 it mention than in this manuscript there is an "exact solution" for the MFPT (that decreases with N). By exact solution, it is meant the numerical solution? Because the analytical solution showing the logarithmic dependence on N is an approximation, according to equation 12.

Reviewer #2: The authors successfully addressed most of my comments and the manuscript has greatly improved.

I have some remaining concerns about the logarithmic scaling of the fixation time. In particular, I have not understood the argument by the authors, according to which the mean fixation time as defined in Koldaeva et al is a different quantity than that studied in this manuscript. I do understand that the initial condition is different: an infinitely diverse initial condition (with each individual belonging to a different strains) in Koldaeva et al. versus two strains involved in a "tug-of-war" here. But this is not really clear from the manuscript, and it would be better to express it in simpler terms.

I also do not think that it is surprising that the log(N) scaling is common to both cases. The common factor is the accelerating dynamics of interfaces far from the center of the channel. This is the case regardless of the initial condition.

Please also note that Koldaeva et al. provides an exact expression for the mean fixation time in the one-lane case. Therefore, the expressions "Using heuristic analysis" at line 83 and "Using simplifying assumptions" at lines 527-528 are not correct.

**Have the authors made all data and (if applicable) computational code underlying the findings in their manuscript fully available?**

Reviewer #1: Yes

Reviewer #2: Yes

PLOS authors have the option to publish the peer review history of their article (what does this mean?). If published, this will include your full peer review and any attached files.

Reviewer #1: No

Reviewer #2: No

Figure Files:

Data Requirements:

Reproducibility:

References:

---

## [Editor Report · Decision Letter 2]

7 Nov 2023

Dear Dr. Zilman,

We are pleased to inform you that your manuscript 'Spatial exclusion leads to tug-of-war ecological dynamics between competing species within microchannels' has been provisionally accepted for publication in PLOS Computational Biology.

Best regards,

Daniel Weissman, Ph.D.

Guest Editor

PLOS Computational Biology

James O'Dwyer

Section Editor

PLOS Computational Biology

The authors have extensively revised and clarified the presentation of their results. I thank them for their hard work! In my opinion, it has paid off in the finished product.

---

## [Editor Report · Acceptance letter]

24 Nov 2023

PCOMPBIOL-D-23-00035R2 

Spatial exclusion leads to tug-of-war ecological dynamics between competing species within microchannels

Dear Dr Zilman,

I am pleased to inform you that your manuscript has been formally accepted for publication in PLOS Computational Biology. Your manuscript is now with our production department and you will be notified of the publication date in due course.

With kind regards,

Zsofia Freund
